# Clinical and Laboratory Characteristics of Anaemia in Hospitalized Patients with Inflammatory Bowel Disease

**DOI:** 10.3390/jcm12072447

**Published:** 2023-03-23

**Authors:** Małgorzata Woźniak, Anna Borkowska, Marta Jastrzębska, Marcin Sochal, Ewa Małecka-Wojciesko, Renata Talar-Wojnarowska

**Affiliations:** 1Department of Digestive Tract Diseases, Medical University of Lodz, 90-419 Lodz, Poland; mmw89@op.pl (M.W.);; 2Department of Internal Diseases and Diabetology, Medical University of Lodz, 90-419 Lodz, Poland; 3Department of Gastroenterology, Health Care Center, 26-200 Konskie, Poland; 4Department of Sleep Medicine and Metabolic Disorders, Medical University of Lodz, 90-419 Lodz, Poland

**Keywords:** inflammatory bowel disease (IBD), ulcerative colitis, Crohn’s disease, anaemia

## Abstract

Anaemia is the most common extraintestinal manifestation of inflammatory bowel disease (IBD). Due to its multifactorial etiopathogenesis, the differential diagnosis and treatment of anaemia in IBD is a significant clinical problem. The main aim of our study was to assess the usefulness of laboratory parameters, including hepcidin, in differential diagnoses of anaemia in hospitalized IBD patients. This study also estimated the impact of anaemia on the length of hospitalization and its relationship with clinical data of analyzed patients. The study included 118 adult patients diagnosed with IBD—55 with ulcerative colitis (UC) and 63 with Crohn’s disease (CD). Anaemia was significantly more frequent in patients with CD—42 (66.7%)—compared to 31 (56.4%) patients with UC (*p* = 0.033). The prevalence of anaemia increased significantly with the severity of IBD and the extent of inflammatory changes in the endoscopic examination. Hospitalization time was significantly longer in patients with anaemia, especially in the group with UC. Ferritin concentrations < 30 ng/mL were found only in 15 (20.55%) IBD patients (9 with UC and 6 with CD), and ferritin < 100 ng/mL was observed in 22 (30.14%) patients, equally frequent with UC and CD (*p* > 0.05). Significantly higher concentrations of transferrin were observed in patients with anaemia in the course of UC compared to CD (2.58 ± 0.90 g/L vs. 2.15 ± 0.82 g/L; *p* = 0.037). On the other hand, saturation of transferrin < 16% was equally common in UC and CD patients. In our study, hepcidin levels in anaemic UC patients were significantly lower compared to UC without anaemia (*p* = 0.042), with no similar differences in CD independently of anaemia presence (*p* = 0.565). To conclude, we observed a high incidence of anaemia in patients with IBD and its significant impact on the length of hospitalization in UC. Routinely determined single laboratory parameters are not sufficient for the differential diagnosis of anaemia, and a complex laboratory assessment, including of hepcidin levels, is necessary for the full picture of anaemia in the course of IBD.

## 1. Introduction

Anaemia is the most common extraintestinal manifestation of inflammatory bowel disease (IBD), i.e., both Crohn’s disease (CD) and ulcerative colitis (UC), and according to the epidemiological data, its prevalence ranges widely from 6% to 74% [1,2,3]. Anaemia has a known significant negative impact not only on health-related quality of life, but also on the ability to work and on cognitive function. The etiology of anaemia in IBD is complex and multifactorial. One of its most common causes is iron deficiency, which affects 36–90% of patients and is provoked by chronic blood loss from inflamed gastrointestinal mucosa or impaired iron absorption [3,4,5]. Anaemia of chronic diseases, resulting from the activation of pro-inflammatory cytokines such as interferon alpha, interleukin 6 and tumor necrosis factor alpha, is also often diagnosed in IBD. These cytokines may cause hepcidin overexpression and, as a result, lead to decreased iron absorption from intestinal cells and the reticulo-epithelial system [6,7]. Other possible causes of anaemia in IBD are vitamin B_12_ or folic acid deficiencies due to malabsorption disorders, resection procedures or the presence of small intestinal bacterial overgrowth [8]. Additionally, anaemia may be caused by side effects of pharmacological treatment, including the use of 5-aminosalicylic acid, thiopurines or methotrexate [9]. In particular, it is known that bone marrow suppression may occur at any time during thiopurines treatment and is associated with thiopurine methyltransferase (TPMT) deficiency [9].

Due to the multifactorial etiopathogenesis, differential diagnosis of anaemia in IBD is a significant clinical problem. The particular challenge is to differentiate anaemia caused by iron deficiency from that of chronic diseases, resulting from the adverse effect of pro-inflammatory cytokines on erythropoiesis. It is well known that the most accurate indicator of iron stores in patients without inflammation is ferritin determined in serum [5]. However, ferritin is an acute phase protein, so its concentration may be normal or elevated in inflammation, even in the presence of iron deficiency. According to the ECCO guidelines, the criterion for the diagnosis of anaemia resulting from iron deficiency in patients without clinical features of IBD exacerbation is a serum ferritin level below 30 µg/L [10]. However, in the presence of inflammation, serum ferritin levels up to 100 µg/L may suggest iron deficiency [11,12]. Another important parameter in the differential diagnosis of anaemia in IBD patients is transferrin saturation (TfS), which is an indicator of iron availability for haematopoiesis. A TfS level of 16–50% is considered to reflect normal iron stores, and a TfS level below 16% corresponds to iron deficiency in the body [11,12,13,14]. In the differential diagnosis of anaemia, the role of hepcidin is also emphasized. This is a hormone that regulates iron metabolism by combining with ferroportin, which controls the outflow of iron from cells. When combined with ferroportin, hepcidin causes its phosphorylation, and as a result, iron is retained in enterocytes, and its removal is promoted along with the process of intestinal epithelium desquamation. On the other hand, hepcidin deficiency leads to increased intestinal absorption of iron and its increased release from macrophages [7]. In one of the recent studies on newly diagnosed IBD patients, Stojkovic Lalosevic et al. suggested that hepcidin can be a reliable marker of iron deficiency anaemia in this group of patients, and it can be used in clinical practice to determinate adequate therapy. The authors showed that serum hepcidin levels were significantly higher in the control group compared to IBD patients (*p* < 0.001) and did not correlate with CRP level (*p* > 0.05) [15]. Therefore, hepcidin is a potentially promising laboratory parameter in the diagnosis of anaemia, although its clinical significance in IBD patients is yet to be confirmed [16].

The main aim of the present study was to assess the usefulness of laboratory parameters, including hepcidin, in the differential diagnosis of anaemia in hospitalized IBD patients. Additionally, this study estimated the impact of anaemia on the length of hospitalization and its relationship with clinical data of patients with CD and UC.

## 2. Materials and Methods

All subsequent IBD patients hospitalized in the Department of Digestive Tract Diseases at the Medical University of Lodz between 2017 and 2019 were qualified for this prospective study. After admission to our department, an additional peripheral blood sample (1 mL) was collected from each patient with IBD who had consented to participate in the study during routine blood sampling for laboratory tests. Depending on the results of their morphology, patients were placed into to the group with or without anaemia. The study excluded patients with a recent history of anaemia treatment, including transfusion of red blood cell concentrate or supplementation with iron, folic acid or vitamin B_12_ within 3 months of the current hospitalization. Exclusion criteria included other accompanying diseases affecting iron management, such as haemochromatosis, porphyria or thalassemia, as well as bone marrow diseases including myelodysplastic syndrome. Moreover, patients with other chronic inflammatory diseases were also excluded from our study. Each patient signed a written consent form to participate in the study. The approval of the Bioethics Committee of the Medical University of Lodz was obtained for the study (nr RNN/149/16/KE).

The study included 118 adult patients diagnosed with IBD, 55 with UC and 63 with CD. We analyzed the clinical data of patients, the length of their hospitalization, as well as the results of endoscopic and laboratory tests. The laboratory parameters included an assessment of haemoglobin (HGB), haematocrit (HCT), red blood cells (RBC), mean corpuscular volume (MCV), mean corpuscular haemoglobin (MCH), mean corpuscular haemoglobin concentration (MCHC), C-reactive protein (CRP), ferritin, transferrin, iron, vitamin B_12_ and hepcidin concentration recorded for each patient. According to the definition of the World Health Organization (WHO), anaemia was defined as HGB < 12 g/dL in non-pregnant women and <13 g/dL in men. Mild anaemia was diagnosed with HGB > 11 g/dL, moderate with HGB 8–11 g/dL, and HGB < 8 g/dL was considered to be severe anaemia [17]. The range of normal values for the other determined parameters were: 70–180 µg/dL for serum iron concentration and 30.0–400.0 ng/mL ferritin, 2.00–3.60 g/L transferrin, vitamin B_12_ 0.4–40 ng/mL vitamin B_12_ and <0.5 mg/L for CRP concentration, respectively. Hepcidin levels were determined by the ELISA method (R&D Systems Company, Minneapolis, MN, USA), and the minimum detected hepcidin level was 0.446 ng/mL.

In the statistical analysis, the results were expressed as mean ± standard deviation. The assumption of the normal distribution of differences was verified with the use of the Shapiro–Wilk test. As the normality assumption was violated, the significance of differences was tested with Mann–Whitney’s U test to compare two independent groups. Inter-group differences of categorical variables, such as gender, were analyzed by chi-square test. *p*-values less than 0.05 were considered statistically significant. All statistical calculations were performed using the Statistica 13.1 program by StatSoft, Inc. (Cracow, Poland).

## 3. Results

During the analyzed time period, 118 IBD patients were qualified to be in the study: 55 (46.61%) with UC and 63 (53.39%) with CD, with no significant differences in terms of age and gender (*p* > 0.05). Anaemia was found significantly more often in patients with CD—42 (66.7%)—compared to 31 (56.4%) patients with UC (*p* = 0.033). The prevalence of anaemia increased significantly with the severity of IBD and the extent of inflammatory changes in the endoscopic examination, both in patients with CD and UC. The clinical data of analyzed patients along with the disease location and endoscopic results are presented in Table 1, Table 2 and Table 3.

The duration of IBD before the current hospitalization was 8.14 ± 6.17 years in patients with CD and 8.13 ± 5.22 years in patients with UC (*p* > 0.05). The average age of IBD patients with anaemia was 43.46 years (40.02 years in patients with CD and 48.13 years in patients with UC) and was similar to the age of patients with IBD without anaemia (*p* > 0.05). Patients with anaemia were significantly more often treated with steroids or azathioprine compared to patients without anaemia (*p* < 0.05). Hospitalization time in patients with IBD was significantly longer in patients with anaemia compared to patients without anaemia, especially in the group of patients with UC (Table 1).

Based on HGB level, severe anaemia occurred in two (2.74%) patients and was diagnosed only in CD patients. Moderate anaemia was most frequently observed, occurring in 44 (60.27%) patients, including 24 (77.42%) patients with UC and 20 (47.62%) with CD (*p* = 0.029). Mild anaemia was diagnosed in 27 patients (36.99%), including 7 (22.58%) patients with UC and 20 (47.62%) with CD (*p* = 0.029; Figure 1). Taking into account other laboratory parameters, normocytic normochromic anaemia was the most common diagnosis in the analyzed IBD group (31 patients, 42.46%), especially in CD (22 patients, 52.38%; Figure 1). The second most frequent laboratory abnormality was microcytic hypochromic anaemia (18 patients, 24.66%), with a predominance of UC (12 patients, 38.71%), followed by normocytic hypochromic anaemia (16 patients both CD and UC, 21.92%; Figure 2).

The average serum iron concentration in IBD patients with anaemia was 30.58 ± 41.28 µg/dL, with no significant differences between patients with UC (29.08 ± 30.83 µg/dL) compared to CD (31.55 ± 47.20 µg/dL, *p* = 0.6; Figure 3). Similarly, there were no differences in average ferritin levels in patients with anaemia and UC or CD (respectively, 280.093 ± 04.39 ng/mL vs. 384.25 ± 329.74 ng/mL; *p* = 0.222). Ferritin concentrations <30 ng/mL were found only in 15 (20.55%) IBD patients (9 with UC and 6 with CD), and ferritin levels <100 ng/mL were observed in 22 (30.14%) patients, equally frequent with UC and CD (*p* > 0.05). In the current study, significantly higher concentrations of transferrin were found in patients with anaemia in the course of UC compared to CD (2.58 ± 0.90 g/L vs. 2.15 ± 0.82 g/L; *p* = 0.037). Additionally, in patients with UC, the levels of transferrin saturation were lower than in patients with CD, but this difference did not reach statistical significance (11.09 ± 10.51% vs. 19.68 ± 19.41%; *p* = 0.09). Transferrin saturation <16%, which according to the ECCO guidelines indicates iron deficiency, was found in 13 patients with UC and 12 patients with CD and anaemia (*p* > 0.05). In our study, vitamin B_12_ below normal values was found only in seven patients with IBD, including three patients with UC and four with CD (*p* > 0.05). Moreover, there was no typical megaloblastic anaemia in the laboratory analysis of blood morphology.

The average concentration of hepcidin in anaemic UC patients was 0.656 ± 0.321 ng/mL and was significantly lower compared to patients with UC without anaemia (0.945 ± 0.449 ng/mL; *p* = 0.042). On the other hand, there were no significant differences in hepcidin levels in CD patients with and without anaemia (1.16 ± 0.946 ng/mL vs. 0.987 ± 0.688; *p* = 0.565; Figure 4). Similarly, no statistically significant differences in hepcidin levels were found, depending on the severity of the disease and the location of changes in both UC and CD (*p* > 0.05).

In the presented study, CRP concentrations in the hospitalized patients with UC was 49.25 ± 78.35 mg/L and was comparable to the CRP levels in patients with CD—54.53 ± 58.60 mg/L (*p* = 0.149). Both UC and CD patients showed a negative correlation between CRP concentration and HGB (Figure 5), which is consistent with the clinical assessment of the study group and the frequent occurrence of anaemia in severe forms of IBD. Moreover, in patients with CD, there was a correlation between iron concentration and CRP in serum (*p* = 0.047), but such a relationship was not observed in patients with UC (*p* = 0.165), which is shown in Figure 6.

## 4. Discussion

In recent years, a steady increase in the incidence of IBD has been observed, with anaemia being the most common extraintestinal symptom of such diseases. In the present study, anaemia was confirmed in 66.7% of patients with CD and 56.4% with UC. The obtained results fit into the wide range of anaemia’s occurrence that is reported in other studies [2,18,19]. In our previously published study regarding patients with newly diagnosed IBD, a high incidence of anemia was also observed [3]. Furthermore, one recently published study showed that anemia in 6.5% of patients preceded the diagnosis of IBD by over a year [20]. On the other hand, Parra et al. showed that only 35.5% of 529 patients with anaemia in the course of IBD had a full laboratory analysis, including the assessment of iron metabolism [21].

In the current study, we confirmed the significant impact of anemia on the patient’s hospitalization time, especially in UC. Similarly, a multi-yearlong observation of a Canadian group of patients showed that the average length of hospitalization in patients with IBD increased in the case of concomitant anemia from 10.1 to 12.2 days [22]. However, in other studies, longer hospitalization times were observed only in patients with CD and anaemia [1,23]. These differences are probably caused by varied study populations and different stages of IBD in the analyzed groups of patients.

Most of the studies published so far confirmed the relationship between the activity of IBD and the presence of anaemia [22,24,25]. Similar results were obtained in the presented study, showing that the incidence of anaemia increases with the severity of the disease and the extension of inflammatory changes in the endoscopic examination, both in patients with CD and UC. We also observed more frequent occurrence of anaemia in CD patients with localization L2 and L3 according to the Montreal classification. In a multicenter Spanish study, anaemia was diagnosed more often in CD patients with fistulas compared to the non-penetrating type of CD with no differences regarding the location of the disease [25]. Lucendo et al., in the group of patients with UC, showed that the percentage of anaemic patients significantly increases with a longer extension of the disease found in the endoscopic examination [25]. The relationship between the occurrence of anaemia and the endoscopic changes, especially visible in UC, suggests that in this group of patients, it is mainly a consequence of overt or occult bleeding from the inflamed mucosa. It was also proved that the presence of anaemia is associated with a worse prognosis, a more severe course of IBD and more frequent complications [24,26,27]. Patients with anaemia often require intensification of IBD treatment. Similarly, our study showed that patients with concomitant anaemia were more often treated with steroids or immunosuppressants. These reports are similar to the results observed in other published research studies [24,27].

In our study, it was shown that the most common type of anaemia in IBD patients was normocytic normochromic anaemia, followed by the microcytic hypochromic type. According to our results, the normocytic normochromic anaemia predominates especially in the group of patients with CD, whereas in patients with UC, microcytic hypochromic anaemia was slightly more common. Similarly, in the study of Lee et al., normocytic normochromic anaemia was also a common finding in patients with CD [28]. Such results may suggest a predominance of anaemia of iron deficiency in UC and of chronic diseases in CD [28,29].

In the current study, no significant differences in the concentration of ferritin were observed in the analyzed groups of IBD patients. As previously mentioned, according to the ECCO guidelines, in inactive IBD, the diagnosis of iron deficiency anaemia is posed on the basis of low serum ferritin levels below 30 ng/mL [10]. In our study, such a concentration of ferritin was observed only in a small group of patients. On the other hand, in the course of active IBD, higher ferritin levels up to 100 ng/mL may also be observed in anaemia caused by iron deficiency. We found that the concentration of ferritin, as an acute phase protein, was elevated in the majority of IBD patients, regardless of the potential aetiology of anaemia. In recent years, the insufficient role of ferritin in the correct diagnosis of iron deficiency anaemia and the significant impact of concomitant inflammation on ferritin have been underlined [14,29]. In a study conducted on a group of over 2000 IBD patients, it was shown that there was no diagnostic threshold for ferritin or transferrin saturation providing sufficient sensitivity and specificity in the diagnosis of iron deficiency [14]. In our study, we also confirmed the data reported by other authors that ferritin is not an optimal marker in the differential diagnosis of anaemia in IBD [12,14,29].

Due to the insufficient sensitivity of other markers in the differential diagnosis of anaemia in IBD, high expectations were associated with the determination of hepcidin levels. Hepcidin is a promising laboratory parameter in the diagnosis of anaemia, although its usefulness in IBD patients is unclear [16]. The present study showed that the hepcidin concentration was significantly lower in patients with UC and anaemia compared to the control group, whereas no such relationship was observed in patients with CD. It is known that the decreased hepcidin level is associated with increased intestinal absorption of iron and its increased release from macrophages. Therefore, a low hepcidin level is a beneficial phenomenon in patients with UC and with anaemia caused by iron deficiency. Other published studies have shown that both patients with UC and CD with concomitant anaemia have lower hepcidin concentrations than in the control group, whereas contrary to our study, no significant differences between the two diseases were found [7,16]. On the other hand, Bergamaschi et al. proved that the concentration of hepcidin increases in patients with IBD and concomitant anaemia of chronic diseases [30]. Moreover, Basseri et al. showed that in a group of CD patients with accompanying anaemia, the concentrations of hepcidin were high and were positively correlated with the levels of interleukin-6 and ferritin [31]. The differences in the obtained results may be related to different criteria for the inclusion of patients in the studies or different methods of hepcidin determination. Moreover, the differences may be also due to changes in hepcidin concentrations between the exacerbation and remission of CD and UC. In the studies presented so far, it was assumed that in anaemia caused by iron deficiency, the concentrations of hepcidin were decreased, and on the contrary, in anaemia due to chronic diseases, they were increased, whereas in anaemia of mixed aetiology, hepcidin levels were reduced or normal [26,29].

It is also obvious that one of the main causes of anaemia in IBD is chronic inflammation. In the presented study, both in CD and UC, a negative correlation was observed between the concentration of CRP and HGB, which confirms the influence of the intensity of inflammation on the incidence of anaemia in IBD. A similar study showed that an increase in CRP of 1 mg/L may increase the risk of anaemia by 80–90% [32]. In patients with CD, in contrast to UC, we also observed a significant relationship between the concentration of CRP and the iron level. This may indicate that the inflammatory component of anaemia is dominant in this group of patients.

One possible cause of anaemia in IBD patients is also vitamin B_12_ deficiency. Although the data from the literature indicate a deficiency of this vitamin in up to 22% of patients with CD [8,33], in the presented study, reduced concentrations of vitamin B_12_ were found in only seven (9.6%) IBD patients, with no significant difference in both CD and UC. From the literature, it is known that in the group of patients with UC, the incidence of vitamin B_12_ deficiency is low and is similar to the general population, with the exception of patients after proctocolectomy [33]. Similarly, in the analyzed study, the incidence of vitamin B_12_ deficiency was low and was not associated with the occurrence of severe anaemia. The isolated deficiency of vitamin B_12_ usually results in megaloblastic anaemia [8,34]. However, in the presented study, in the group of patients with CD with vitamin B_12_ deficiency, normocytic normochromic anaemia prevailed, whereas in the group of patients with UC, despite vitamin B_12_ deficiency, hypochromic microcytic anaemia dominated. The above results may suggest that vitamin B_12_ deficiency in patients with IBD coexists with another cause of anaemia, and the typical picture of megaloblastic anaemia is extremely rare in this group of patients.

A clear strength of our study is the availability of the analyzed parameters that can be easily measured in clinical practice, not only in specialized medical centers. The accessibility of technical instruments and the low cost of laboratory tests allow them to be potentially widely used for the differential diagnosis of anemia in the course of IBD. On the other hand, a possible limitation of this study is the relatively small and clinically diverse group of analyzed IBD patients. However, it is highly unlikely that we missed a significant association between clinical data and analyzed laboratory tests. Nevertheless, it is sure that the studied issue deserves further research.

## 5. Conclusions

In conclusion, in our study, we observed a high incidence of anaemia in patients with IBD and noted its impact on the length of hospitalization in UC. Routinely determined single laboratory parameters, including ferritin, are not sufficient for the differential diagnosis of anaemia in IBD because of its multifactorial mechanisms. A complex laboratory assessment, including the determination of hepcidin and transferrin saturation, is important for the full picture of anaemia in the course of IBD. It is certain that due to its clinical significance, this problem is worthy of further analysis and research studies.

## Figures and Tables

**Figure 1 jcm-12-02447-f001:**
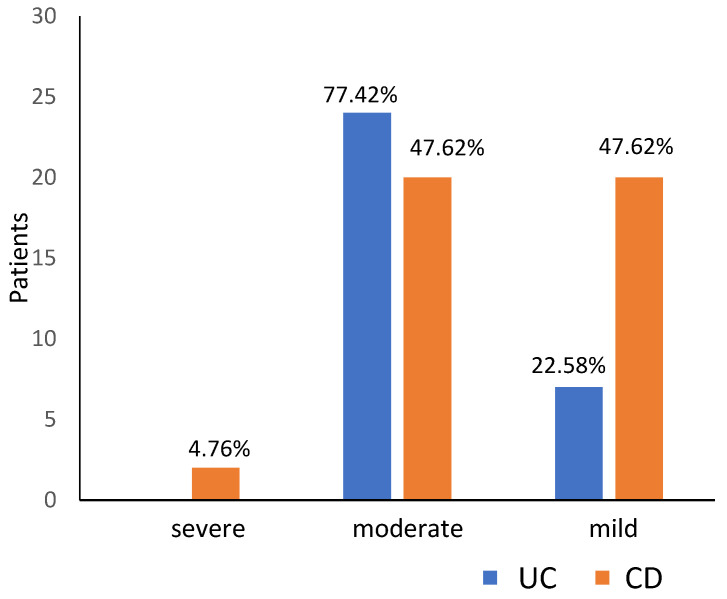
Severity of anaemia in patients with ulcerative colitis (UC) and Crohn’s disease (CD).

**Figure 2 jcm-12-02447-f002:**
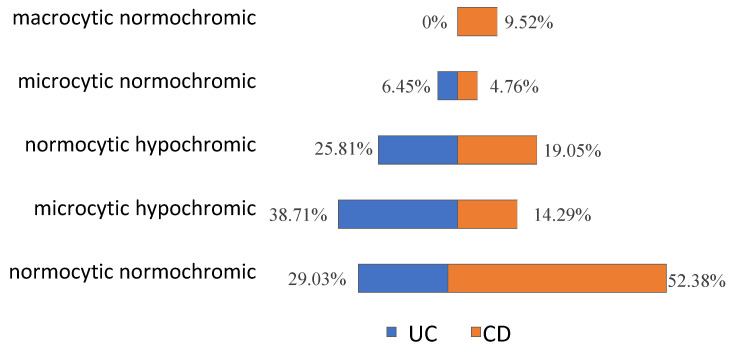
The prevalence of particular types of anaemia in patients with ulcerative colitis (UC) and Crohn’s disease (CD).

**Figure 3 jcm-12-02447-f003:**
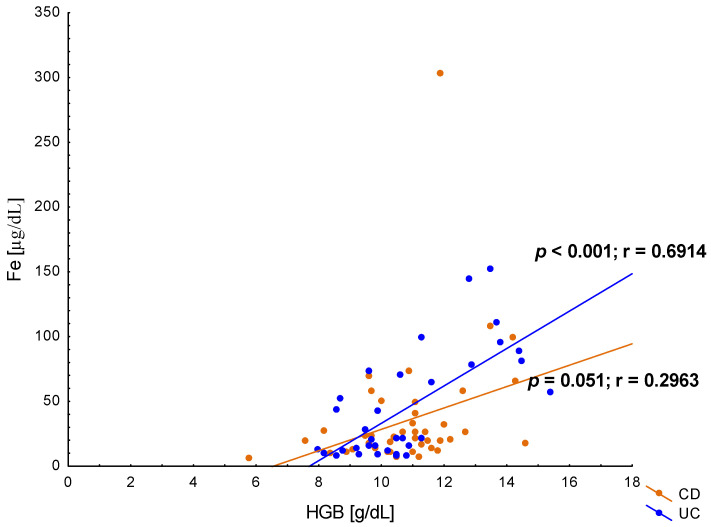
The correlation between iron (Fe) and haemoglobin (HGB) concentration in patients with ulcerative colitis (UC) and Crohn’s disease (CD).

**Figure 4 jcm-12-02447-f004:**
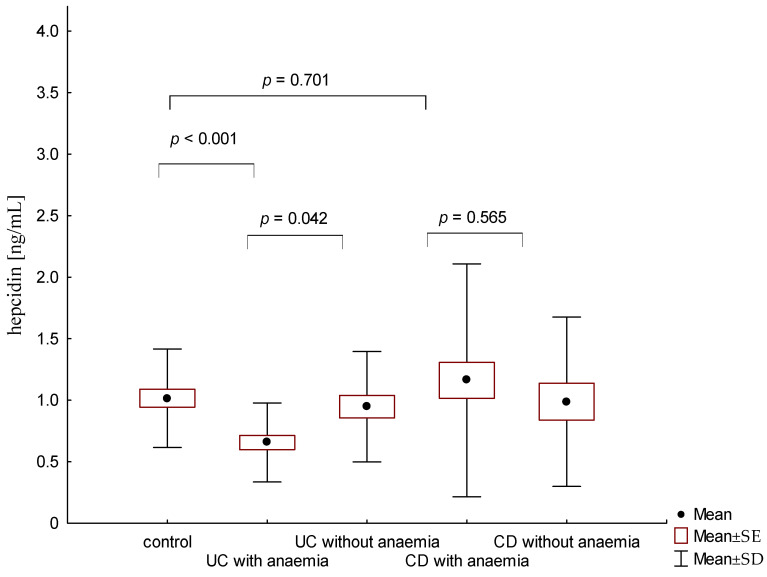
The concentration of hepcidin in patients with ulcerative colitis (UC) and Crohn’s disease (CD) with or without anaemia in relation to the control group.

**Figure 5 jcm-12-02447-f005:**
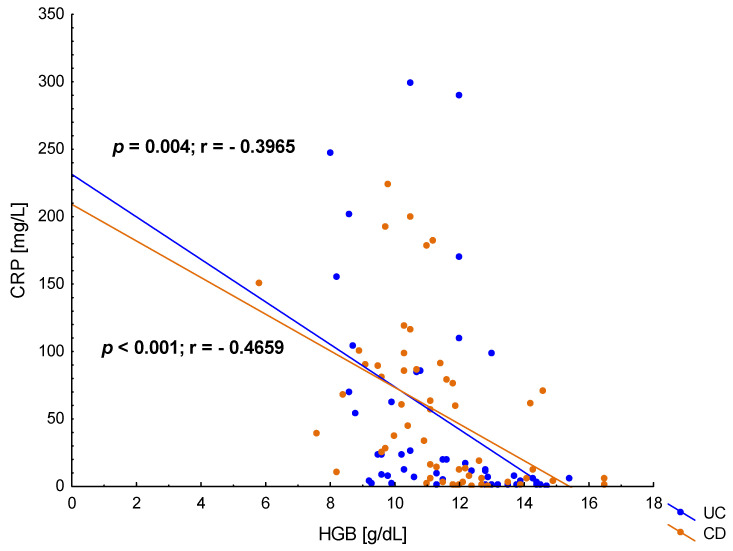
The correlation between C-reactive protein (CRP) and haemoglobin (HGB) concentration in patients with ulcerative colitis (UC) and Crohn’s disease (CD).

**Figure 6 jcm-12-02447-f006:**
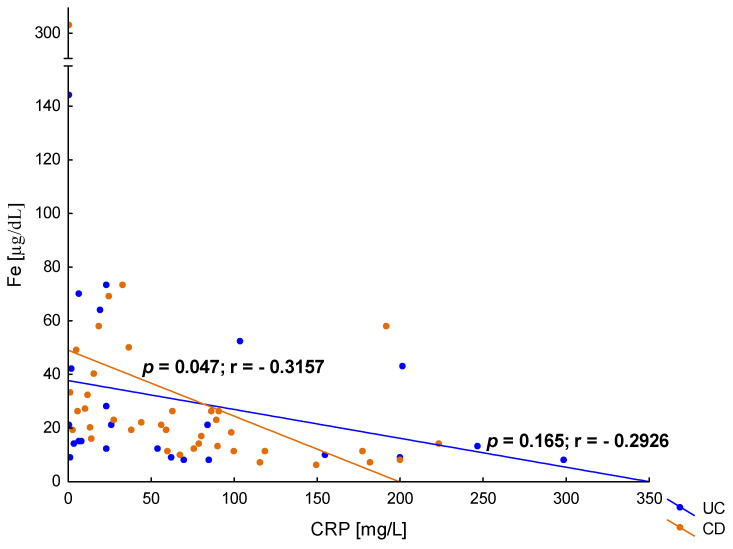
The correlation between iron (Fe) and C-reactive protein (CRP) concentration in patients with ulcerative colitis (UC) and Crohn’s disease (CD).

**Table 1 jcm-12-02447-t001:** Characteristics of the examined patients with IBD.

		IBD			*p* *	CD			*p* *		UC		*p* *
		Overall	Anaemia	Without Anaemia		Overall	Anaemia	Without Anaemia		Overall	Anaemia	Without Anaemia	
N		118	73	45	**<0.01**	63	42	21	**<0.01**	55	31	24	0.09
	men	51	34	17	**<0.01**	27	19	8	**<0.01**	24	15	9	**0.04**
	women	67	39	28	**0.03**	36	23	13	0.01	38	16	15	0.40
Age (years)		42,5 (15,6)	41,4 (16,5)	40,9 (12,5)	0.70	38,7 (14,1)	40,0 (15,9)	36,2 (9,5)	0.67	46,8 (16,2)	43,0 (17,2)	45,1 (13,5)	0.45
(SD)	men	42,1 (14,8)	44,5 (17,1)	37,2 (6,9)	0.17	38,8 (13,7)	40,2 (15,7)	35,4 (6,4)	0.71	45,8 (15,4)	50,0 (17,6)	38,9 (7,4)	0.09
	women	42,8 (16,2)	38,7 (14,6)	43,2 (14,6)	0.19	38,7 (14,6)	39,9 (16,3)	36,7 (11,2)	0.79	47,6 (16,9)	46,4 (18,8)	48,1 (15,1)	0.68
Lenght of		7,33 (4,8)	8,00 (4,9)	3,67 (2,4)	**<0.01**	6,64 (3,8)	6,79 (3,7)	4,0 (2,4)	0.08	8,27 (5,9)	9,58 (5,9)	3,43 (2,6)	**<0.01**
Hospitalisati	men	7,41 (5,6)	8,45 (5,8)	3,38 (2,4)	**<0.01**	6,95 (4,5)	7,68 (4,4)	2,33 (1,1)	**0.02**	8,00 (6,9)	9,67 (7,5)	4,00 (2,8)	**0.04**
(days) (SD)	women	7,26 (3,9)	7,60 (3,9)	4,25 (2,7)	0.10	6,35 (2,98)	6,33 (3,1)	6,5 (0,7)	0.66	8,56 (4,8)	9,50 (4,4)	2,00 (1,4)	**0.04**

* comparing anaemia and without anaemia.

**Table 2 jcm-12-02447-t002:** Clinical characteristics of patients with CD with and without anaemia.

		CD			*p* *
		Overall n = 63	Anaemia n = 42	Without Anaemia n = 21	
disease location	L1	16 (27,59)	6 (15,4)	10 (52,6)	0.07
(%)	L2	17 (29,31)	13 (33,3)	4 (21,1)	**<0.01**
	L3	25 (43,10)	20 (51,3)	5 (26,3)	**<0.01**
	L4	0 (0)	0 (0)	0 (0)	-
disease	B1	32 (54,24)	22 (55,0)	10 (51,6)	**<0.01**
behaviour(%)	B2	6 (10,17)	3 (7,5)	3 (15,8)	0.50
	B3	21 (35,59)	15 (37,5)	6 (31,6)	**<0.01**
age at onset	A1	2 (3,39)	2 (5,1)	0 (0,0)	**<0.01**
(%)	A2	43 (72,88)	25 (64,1)	18 (90,0)	0.06
	A2	14(23,73)	12 (30,8)	2 (10,0)	**<0.01**
CDAI (%)	remission	11 (18,03)	1 (2,4)	10 (50,0)	**<0.01**
	mild	8 (13,11)	4 (9,8)	4 (20,0)	0.05
	moderate	22 (36,07)	16 (30,0)	6 (30,0)	**<0.01**
	severe	20 (32,79)	20 (48,8)	0 (0,0)	**<0.01**
SES-CD (%)	remission	12 (21,82)	4 (10,8)	8 (44,4)	**0.04**
	mild	10 (18,18)	6 (16,2)	4 (22,2)	0.18
	moderate	17 (30,91)	12 (32,4)	5 (5,6)	**<0.05**
	severe	16 (29,09)	15 (40,6)	1 (27,8)	**<0.01**

L—disease location: L1—ileal disease; L2—colonic disease; L3—ileocolonic disease; L4—upper gastrointestinal tract; B-disease behaviour: B1-non-stricturing and non-penetrating type, B2-stricturing disease, B3-penetrating disease; A—age at onset: A1—≤16 yr, A2—17–39 yr, A3—≥40 yr; * comparing anaemia and without anaemia.

**Table 3 jcm-12-02447-t003:** Clinical characteristics of patients with UC with and without anaemia.

		UC			*p* *
		Overall n = 55	Anaemia n = 31	Without Anaemia n = 24	
Extent of disease	E1	6 (11,3)	5 (16,7)	1 (4,3)	**<0.01**
(%)	E2	10 (18,9)	8 (26,7)	2 (8,7)	**<0.01**
	E3	18 (34,0)	14 (46,6)	4(17,4)	**<0.01**
	remission	19 (35,9)	3 (10,0)	16 (69,6)	**<0.01**
Mayo (%)	remission	9 (16,7)	0 (0,0)	9 (39,1)	**<0.01**
	mild	13 (24,1)	5 (16,2)	8 (34,8)	0.11
	moderate	18 (33,3)	13 (41,9)	5 (21,7)	**<0.05**
	severe	14 (25,9)	13 (41,9)	1 (4,4)	**<0.01**
Mayo	Mayo 0	19 (35,2)	3 (9,7)	16 (69,6)	**<0.01**
Endoscopic (%)	Mayo 1	6 (11,1)	4 (12,9)	2 (8,7)	0.11
	Mayo 2	14 (25,9)	11 (35,5)	3 (13,0)	**<0.01**
	Mayo 3	15 (27,8)	13 (41,9)	2 (8,7)	**<0.01**

E—disease extension: E1—proctitis, E2—left-sided UC, E3—extensive (pancolitis) UC; * comparing anaemia and without anaemia.

## Data Availability

The data presented in this study are available on request from the corresponding author.

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
