# Peer review of "Clinical and Laboratory Characteristics of Anaemia in Hospitalized Patients with Inflammatory Bowel Disease"

_jcm, 2023, doi:10.3390/jcm12072447_

Round 1

Reviewer 1 Report

This is an overall well-written manuscript. The authors present a study with 118 patients with inflammatory bowel disease. Among them, 55 patients are with ulcerative colitis (UC) and 63 are with Crohn’s disease (CD). The objective of the study is to assess the usefulness of laboratory parameters in differential diagnosis of anaemia in hospitalized IBD patients. A second objective is to assess the impact of anaemia on the length of hospitalization 80 and its relationship with clinical data of patients with CD and UC. The objectives are presented clearly with support from the results, conclusion and discussion. The study measurements and statistical methods are reasonable. The study helps understand anaemia in IBD and it is important for future clinical research. I have the following comments that I hope the authors could address:

1.     The way Table 1 and Table 2 are organized with many blanks and repetitive information is not very useful and helpful for readers to review. They could be improved and reorganized. I suggest to include one characteristics table where data are all available for IBD (with anemia, without anemia, overall), CD (with anemia, without anemia, overall), UC (with anemia, without anemia, overall). Then the authors could use two separate tables for CD and UC each, with the disease specific characteristics. In addition, other socio-demographic variables and clinical outcomes are of interest. If they are collected, please provide accordingly.

2.     Statistical comparisons and p-values might be overly used. Especially when there are multiple group comparisons. Please consider to acknowledge the risks of type I error and describe efforts to mitigate this issue. In addition, please justify which comparisons and p-values are from t-student test, Mann-Whitney U test or chi-squared test, and how are they chosen. Because they each have specific data assumption requirements.

3.     Minor format issue: comma sign “,” is used instead of period sign “.” in the figures, tables and several places in the body. For example, page 6 line 161 “p=0,6” and all tables/figures. However, other places in the main body used period sign “.” correctly. It would be nice if they could be consistent and follow international standard.

Author Response

We would like to thank the Editor and the Reviewers for the thorough and critical reviews. We modified the manuscript according to the Reviewers’ suggestions and we hope it helped to enhance our work to be acceptable to publication in Journal of Clinical Medicine.

Specific comments:

  1. The way Table 1 and Table 2 are organized with many blanks and repetitive information is not very useful and helpful for readers to review. They could be improved and reorganized. I suggest to include one characteristics table where data are all available for IBD (with anemia, without anemia, overall), CD (with anemia, without anemia, overall), UC (with anemia, without anemia, overall). Then the authors could use two separate tables for CD and UC each, with the disease specific characteristics. In addition, other socio-demographic variables and clinical outcomes are of interest. If they are collected, please provide accordingly.

Thank you very much for this comment. We modified both Tables according suggestions to make it more clear for the reader.

  1. Statistical comparisons and p-values might be overly used. Especially when there are multiple group comparisons. Please consider to acknowledge the risks of type I error and describe efforts to mitigate this issue. In addition, please justify which comparisons and p-values are from t-student test, Mann-Whitney U test or chi-squared test, and how are they chosen. Because they each have specific data assumption requirements.

Thank you very much for this comment. In the statistical analysis the results were expressed as mean +/- standard deviation. Assumption of the normal distribution of differences was verified with the use of the Shapiro–Wilk test. As the normality assumption was violated, the significance of differences was tested with Mann–Whitney’s U test to compare two independent groups. Inter-group differences of categorical variables, such as gender, were analyzed by chi-square test. P-values less than 0.05 were considered statistically significant. All statistical calculations were performed using the Statistica program by StatSoft, Inc (Poland). This explanation was added into the text.

  1. Minor format issue: comma sign “,” is used instead of period sign “.” in the figures, tables and several places in the body. For example, page 6 line 161 “p=0,6” and all tables/figures. However, other places in the main body used period sign “.” correctly. It would be nice if they could be consistent and follow international standard.

Thank you very much for pointing this out. We corrected all the manuscript according to international standard.

Reviewer 2 Report

The main aim of our study was to assess the usefulness of laboratory parameters, including hepcidin, in the differential diagnosis of anemia in hospitalized IBD patients.

Major Comments:

1.     It was recently shown by Milica Stojkovic Lalosevic et. al., that Hepcidin Is a reliable biomarker of Iron Deficiency Anemia in newly diagnosed Inflammatory Bowel Disease patients. This should be discussed in the introduction section of the paper.

2.     On line 47, the authors state that – “these cytokines may cause hepcidin overexpression and, as a result, lead to decreased iron absorption from intestinal cells and the reticulo-epithelial system [6,7]”. The authors cite the review which mentioned this but they do NOT cite the actual manuscript which demonstrated this for the first time. It is important to give due credit to the scientists who made this discovery through the citation of their work.

3.     The authors must perform a ROC curve analysis for hepcidin in IBD patients (UC and CD).

Minor Comments:

1.     On line 186, the authors state that – “In the presented study, CRP concentrations in the hospitalized patients with UC was 49.25±78.35 mg/l and was comparable to the CPR levels in patients with CD–54.53± 58.60 mg/l (p=0.149).” Here, instead of CPR, it should be CRP. This mistake is repeated in line 191.

Author Response

We would like to thank the Editor and the Reviewers for the thorough and critical reviews. We modified the manuscript according to the Reviewers’ suggestions and we hope it helped to enhance our work to be acceptable to publication in Journal of Clinical Medicine.

Specific comments:

  1. It was recently shown by Milica Stojkovic Lalosevic et. al., that Hepcidin Is a reliable biomarker of Iron Deficiency Anemia in newly diagnosed Inflammatory Bowel Disease patients. This should be discussed in the introduction section of the paper.

Thank you very much for this suggestion. We analyzed this article and discussed their results in the introduction section of the manuscript.

  1. On line 47, the authors state that – “these cytokines may cause hepcidin overexpression and, as a result, lead to decreased iron absorption from intestinal cells and the reticulo-epithelial system [6,7]”. The authors cite the review which mentioned this but they do NOT cite the actual manuscript which demonstrated this for the first time. It is important to give due credit to the scientists who made this discovery through the citation of their work.

Thank you very much for pointing this out. We corrected manuscript according this suggestions. We cited two articles:

1.Wang, L.;Trebicka, E.; Fu, Y.;Ellenbogen, S.; Hong, C.C.; Babitt, J.L.; Lin, H.Y.,Cherayil, B.J. The bone morphogenetic protein-hepcidin axis as a therapeutic target in inflammatory bowel disease. Inflamm Bowel Dis2012;18(1), 112-9.

2.Shu, W.; Pang, Z.; Xu, C.; Lin, J.; Li, G.; Wu, W.; Sun, S.; Li, J.; Li, X.; Liu, Z. Anti-TNF-α Monoclonal Antibody Therapy Improves Anemia through Downregulating Hepatocyte Hepcidin Expression in Inflammatory Bowel Disease. Mediators Inflamm2019, 2019:4038619.

3.The authors must perform a ROC curve analysis for hepcidin in IBD patients (UC and CD).

Thank you very much for this suggestion, but unfortunately ROC curve analysis for hepcidin in our study was not significant.

  1. On line 186, the authors state that – “In the presented study, CRP concentrations in the hospitalized patients with UC was 49.25±78.35 mg/l and was comparable to the CPR levels in patients with CD–54.53± 58.60 mg/l (p=0.149).” Here, instead of CPR, it should be CRP. This mistake is repeated in line 191.

Thank you very much for pointing this out. We checked the manuscript and corrected all mistakes in writing “CRP”.

Reviewer 3 Report

This study compared laboratory parameters and assessed the impact of anaemia on the length of hospitalization  in hospitalized IBD patients. Although the results of the study did not provide  clear differential diagnostic criteria, it  is still an interesting article because of the significance of IBD. However, I have a fewer suggestions for authors to consider.

 1.The study excluded patients with a recent history of anaemia treatment, but did not consider the effects of other anemia-related diseases. The inclusion criteria can be further improved, 

2.The authors used several analysis methods,including t-student test, Mann-Whitney U test and χ2 test.However, it is not clear which analysis each method was  used for.

3. The authors should discuss  why the results were contrary to the previous research, whether it was due to different detection methods or other reasons.

4. A discussion of the strengths and limitations of the study is needed.

Author Response

We would like to thank the Editor and the Reviewers for the thorough and critical reviews. We modified the manuscript according to the Reviewers’ suggestions and we hope it helped to enhance our work to be acceptable to publication in Journal of Clinical Medicine.

Specific comments:

1.The study excluded patients with a recent history of anaemia treatment, but did not consider the effects of other anemia-related diseases. The inclusion criteria can be further improved. 

Thank you very much for this comment. Exclusion criteria included other accompanying diseases affected iron management, such as haemochromatosis, porphyria or thalassemia as well as bone marrow diseases including myelodysplastic syndrome. Moreover, patients with other chronic inflammatory diseases were also excluded from our study. This informations were added into the methodology section in the manuscript.

2.The authors used several analysis methods,including t-student test, Mann-Whitney U test and χ2 test.However, it is not clear which analysis each method was  used for.

Thank you very much for this comment. In the statistical analysis the results were expressed as mean +/- standard deviation. Assumption of the normal distribution of differences was verified with the use of the Shapiro–Wilk test. As the normality assumption was violated, the significance of differences was tested with Mann–Whitney’s U test to compare two independent groups. Inter-group differences of categorical variables, such as gender, were analyzed by chi-square test. P-values less than 0.05 were considered statistically significant. All statistical calculations were performed using the Statistica program by StatSoft, Inc (Poland). This explanation was added into the text.

  1. The authors should discuss  why the results were contrary to the previous research, whether it was due to different detection methods or other reasons.

Thank you very much for this comment. The differences in the obtained results may be related to different criteria for inclusion of patients in the studies or different methods of hepcidin determination. Moreover, the differences may be also due to changes in hepcidin concentrations between the exacerbation and remission of CD and UC.

  1. A discussion of the strengths and limitations of the study is needed.

Thank you very much for pointing this out. Clear strength of our study is the availability of the analyzed parameters that can be easily measured in clinical practice, not only in specialized medical centers. The accessibility of technical instruments and a low cost of laboratory tests make them potentially widely used for differential diagnosis of anemia in the course  of IBD. On the other hand, a possible limitation of this study is relatively small and clinically diverse group of analyzed IBD patients. However, it is highly unlikely that we missed a significant association between clinical data and analyzed laboratory tests. Nevertheless, it is sure that the studied issue deserves further researches.

Round 2

Reviewer 2 Report

Authors have addressed my comments.